# Genome Variability of Infectious Bronchitis Virus in Mexico: High Lineage Diversity and Recurrent Recombination

**DOI:** 10.3390/v15071581

**Published:** 2023-07-20

**Authors:** Ana Marandino, Lizbeth Mendoza-González, Yanina Panzera, Gonzalo Tomás, Joaquín Williman, Claudia Techera, Amanda Gayosso-Vázquez, Vianey Ramírez-Andoney, Rogelio Alonso-Morales, Mauricio Realpe-Quintero, Ruben Pérez

**Affiliations:** 1Sección Genética Evolutiva, Departamento de Biología Animal, Instituto de Biología, Facultad de Ciencias, Universidad de la República, Iguá 4225, Montevideo 11400, Uruguay; amarandino@fcien.edu.uy (A.M.); ypanzera@fcien.edu.uy (Y.P.); gtomas@fcien.edu.uy (G.T.); jwilliman@fcien.edu.uy (J.W.); ctechera@fcien.edu.uy (C.T.); 2Centro Universitario de Ciencias Biológicas y Agropecuarías, Universidad de Guadalajara, Zapopan 44600, JAL, Mexico; liichi1614@gmail.com; 3Departamento de Genética y Bioestadística, Facultad de Medicina Veterinaria y Zootecnia, Universidad Nacional Autónoma de México, Ciudad Universitaria, Ciudad de México 04510, CP, Mexico; amandagv66@hotmail.com (A.G.-V.); vianny102@hotmail.com (V.R.-A.); ralonsom@unam.mx (R.A.-M.)

**Keywords:** IBV, Mexico, genomes, lineages

## Abstract

The avian infectious bronchitis virus (IBV) is a coronavirus that mutates frequently, leading to a contagious and acute disease that results in economic losses to the global poultry industry. Due to its genetic and serological diversity, IBV poses a challenge in preventing and controlling the pathogen. The full-length S1 sequence analysis identifies seven main genotypes (GI–GVII) comprising 35 viral lineages. In addition to the previously described lineage, a new GI lineage (GI-30) and two lineages from novel genotypes (GVIII-1 and GIX-1) have been described in Mexico. To prevent the spread of IBV outbreaks in a specific geographic location and select the suitable vaccine, it is helpful to genetically identify the circulating IBV types. Moreover, sequencing genomes can provide essential insights into virus evolution and significantly enhance our understanding of IBV variability. However, only genomes of previously described lineages (GI-1, GI-9, GI-13, and GI-17) have been reported for Mexican strains. Here, we sequenced new genomes from Mexican lineages, including the indigenous GI-30, GVIII-1, and GIX-1 lineages. Comparative genomics reveals that Mexico has relatively homogenous lineages (i.e., GI-13), some with greater variability (i.e., GI-1 and GI-9), and others extremely divergent (GI-30, GVIII-1, and GIX-1). The circulating lineages and intra-lineage variability support the unique diversity and dynamic of Mexican IBV.

## 1. Introduction

Coronaviruses, which belong to the family Coronaviridae and subfamily Orthocoronavirinae, can cause severe respiratory and digestive illnesses in birds and mammals. The subfamily comprises four genera that exhibit specific host preferences. Delta and Gammacoronavirus are associated with birds, while Alpha and Betacoronavirus are associated with mammals. The first discovered coronavirus was the avian gammacoronavirus infectious bronchitis virus (IBV, later included in the *Avian coronavirus* species) (International Committee on Taxonomy of Viruses, http://www.ictvonline.org/virustaxonomy.asp, accessed on 15 February 2023). It is the most important and best-studied *Gammacoronavirus*, serving as the prototype of the genus, and is found worldwide in industrial and backyard chickens [1,2,3]. IBV was identified in the first third of the 20th century as the causative agent of an acute and highly contagious chicken disease, posing a significant economic burden on the poultry industry [4]. The virus affects the upper respiratory and reproductive tracts; some strains cause nephritis and enteritis. 

The IBV genome is a single-stranded positive-sense RNA molecule of nearly 27 kb surrounded by a lipid envelope. The multicistronic genome comprises two untranslated regions at the 5′ and 3′ ends and open reading frames (ORFs) that code for structural and nonstructural proteins [5]. 

On the IBV surface are projections formed by the membrane spike glycoprotein, which consists of approximately 1145 amino acids [6,7]. This protein is responsible for the binding and entry of the virus into host cells, and it also induces the production of neutralizing and serotype-specific antibodies [8,9,10]. Proteases cleave the precursor S protein into two polypeptides, S1 and S2, which are non-covalently bound. S1, approximately 535 amino acids, form the spike’s tip, while S2, approximately 627 amino acids, anchors S1 into the viral membrane.

IBV has an extraordinary genetic and serological diversity, presenting challenges for prevention and control. The genetic diversity of IBV is generated by single nucleotide substitutions, insertions, deletions, and recombination due to the inaccuracy and random template switching of the coronavirus RNA-dependent RNA polymerase during viral replication [3]. Thereby, variant strains and new genotypes of IBV are continuously emerging worldwide.

In contrast to coronaviruses that affect mammals, which occur as only one or two different serotypes, IBV has many different serotypes with poor cross-protection [11]. Attenuated live vaccines are used in broilers and pullets, and killed vaccines are used in layers and breeders. An effective control strategy involves identifying the virus type causing outbreaks followed by vaccination with an appropriate vaccine. However, only a few IBV vaccines are available in contrast with the numerous types and variants of the virus worldwide. Using a vaccine with unmatching circulating strains offers less protection and might be risky because live vaccines recombine with field strains and directly impact viral evolution [12,13,14,15].

Knowing the genotype or serotype of strains is crucial for effective disease prevention due to the IBV antigenic variability and the lack of cross-protection from commercial vaccines. However, the need for standard positive serum makes identifying serotypes challenging and time-consuming. Thus, genotype classification based on the S1 coding region becomes the primary method for classifying IBV strains [7,9,16]. 

A thorough analysis of the S1 region has led to the identification of six major genotypes (GI-GVI), 32 viral lineages (numbered 1 to 32), as well as a few inter-lineage recombinants found in worldwide strains [17]. This classification system initially consisted of twenty-seven lineages for Genotype 1 and one for the other five genotypes. However, the number of viral types has since increased over the years, with new lineages being discovered for Genotype 1 (GI-28 and GI-29) and the emergence of a new GVII genotype (GVII-1) [18,19,20]. 

Infectious bronchitis is a prevalent respiratory disease among Mexican poultry. Even vaccinated chickens may face challenges with the virus, despite using different vaccine strains, causing a tremendous economic impact. Unfortunately, the genetic characterization of IBV from Mexico is still limited.

Two Mexican complete S1 sequences were included in the S1 classification [17]. The 98-07484 (AF288467) strain was denoted as a unique variant (UV), and the BL-56 strain (AF352831) was classified within the GI-3 lineage. The GI-1, GI-9, GI-13, and GI-17 lineages have recently been reported for Mexico [21,22]. Complete genomes of these GI lineages were obtained and analyzed, confirming the similarity with strains circulating on other continents and evidencing the existence of recombinants [22]. Furthermore, in addition to the previously described lineage, a new GI lineage (GI-30) and two lineages from novel genotypes (GVIII-1 and GIX-1) have been described in Mexico [21]. The new GVIII-1 includes two strains clustered with the unique 98-07484 Mexican variant (AF288467 or UNAM-97) previously described in the S1 classification [17]. Although the S1 sequence is available for the novel GI-30, GVIII-1, and GIX-1 lineages, the rest of the genome is unknown. Genome sequencing aids virus research and tracking and identifies genetic markers for pathogenicity and vaccine development.

In the Mexican poultry industry, live and inactivated vaccines are used to control infectious bronchitis. The officially authorized live virus vaccines are of the Massachusetts, Connecticut, and 793B types.

The present study focuses on investigating complete genomes of IBV strains belonging to previously characterized genotypes (GI-3, GI-9, and GI-13) and novel lineages (GI-30) and genotypes (GVIII-1 and GIX-1) in commercial poultry farms in Mexico. Our findings provide information on the genomic evolution of IBV, which will benefit disease surveillance and outbreak management in the Mexican poultry industry.

## 2. Materials and Methods

### 2.1. Samples

The strains were obtained from archive samples. Seven samples were collected for different outbreaks in commercial broilers with respiratory signs during 2018–2021 in Jalisco, Mexico. Seven samples were isolated from the Universidad Nacional Autónoma de México (UNAM) collected in 2007 from Central Mexico (Table 1). Strains were previously classified by Sanger sequencing of the S1 region [21]. 

### 2.2. Propagation and Purification of IBV Particles

IBV virions were obtained by inoculating strains in embryonated chicken eggs and harvesting allantoic fluid after 72 h. The fluid was concentrated and filtered using 0.45 µm membrane filters for purification.

### 2.3. RNA Extraction and Illumina Sequencing

Viral RNA was extracted with Quick-RNA^TM^ MiniPrep kit (Zymo Research, Irvine, CA, USA) and converted to double-stranded cDNA with Maxima H Minus kit using random primers (Thermo Fisher Scientific, Waltham, MA, USA). Nextera™ DNA Flex Library Preparation kit (Illumina, San Diego, CA, USA) was used for library preparation. Purified libraries were quantified with Qubit (Thermo Fisher Scientific, Waltham, MA, USA) and sequenced on the MiniSeq platform (Illumina, San Diego, CA, USA).

### 2.4. Genome Assembly and Annotation

The BBDuK and Minimap2 plugins in Geneious Prime 2020.1.2 (https://www.geneious.com, accessed on 15 November 2022) were used to trim and filter the raw data and map the clean reads to reference genomes. Assemblies were visually inspected and manually optimized to obtain a single contig; annotations were transferred from reference strains using Geneious. The sequences were deposited in the GenBank database; accession numbers are listed in Table 1.

### 2.5. Genome Sequence Analysis 

A comprehensive genome dataset was compiled using global reference strains representing all lineages with available full-length sequences. 

DNA alignments were performed with MAFFT [23]. The optimal nucleotide substitution model was chosen using the Akaike and Bayesian information criteria in jModelTest.

SplitsTree5 v 5.0.0_alpha was used to determine the likelihood of recombination events in the complete genome sequences of all Mexican strains.

PhyML was used to infer maximum-likelihood trees which were then subjected to approximate likelihood ratio tests for internal node support. The resulting trees were visualized using the iTOL v4 online tool (https://itol.embl.de/, accessed on 15 November 2022). To identify potential recombinant and parental sequences and localize possible recombinant breakpoints, we utilized the RDP4 program (RDP v.4.97) [24].

We employed seven methods (RDP, GENECONV, BootScan, MaxChi, Chimaera, SiScan, and 3Seq) to explore the putative recombination events. Recombination events were positive when supported by six methods with a *p*-value adjusted to 0.05. In addition, the potential recombination events were further verified by phylogenetic incongruence in RDP.

## 3. Results

### 3.1. Lineage Assignment 

A full-length S1 dataset was assembled using sequences retrieved from the 14 genomes here obtained (Table 1), from the other 33 Mexican genomes available [22], from prototype IBV strains (*n* = 32) [17], and from additional viruses from recently identified lineages and genotypes: five strains for GI-28 [25], three for GI-29 [18], and two for GVII-1 [19] (Figure 1). 

Phylogenetic clustering confirms the previous lineage characterization (Figure 1). Ten samples belonged to four lineages of Genotype 1 (G1-3, GI-9, GI-13, and GI-30). The remaining four samples belong to the first described lineage of Genotype VIII (GVIII-1) or GIX (GIX-1) (22). The other 33 Mexican strains’ classification coincided with their previous lineage classification [22]. 

### 3.2. Genome Sequencing and Variability

The Illumina sequencing generated about 1 × 10^5^ reads for each of the fourteen samples; a high percentage of these reads (~70%) matched with IBV strains. The reads were assembled using reference sequences, and the most similar reference was selected to improve the mapping to obtain the genomes of all the strains. 

Differences in the coding sequences were caused by small indels (insertion/deletion) of three or multiples of three nucleotides or alternative codons in the E, 4b, and 4c ORFs. The GI-3 and the Mex-56-7 (GIX-1) strains have a premature stop in the ORF-6b, suggesting that this ORF may be absent. 

### 3.3. Comparison of Complete Genomes

The relationship between 47 complete Mexican genomes was explored using network analysis (Figure 2). The analysis suggests the occurrence of reticulate evolutionary events, such as recombination, indicated by the presence of reticulated nodes with multiple parents. 

The network suggests recombination events both between and within lineages. For example, lineages like GI-3 and GVIII-1 were grouped in the network, while strains from GI-9, GI-30, and GIX-1 were located at different positions. 

Mex-1 (GI-30) formed a split with strains GI-9 and GI-13/GI-17, while Mex-56-7 (GIX-1) was associated with GI-3 strains and Mex-14 (GIX-1) with GVIII-1 strains.

We verified the putative recombination and breakpoint events by analyzing each gene phylogenetically and with the RDP4 program. Further phylogenetic analyses with separate genes confirmed that Mexican lineages are highly variable throughout their genomes and that recurrent recombination occurs (Appendix A).

**GI-3 (JMK/Gray strain-like).** In the phylogenies based on genes 1, 3, 4, 5, and 6, the Mexican strains sequenced here were associated in a single clade. However, the other two previously sequenced GI-3 Mexican strains and the GI-3 referent genomes formed independent clades. In the gene 2 phylogeny, all Mexican strains were associated in a single clade. 

**GI-9 (Ark strain-like).** In the phylogenies based on genes 1, 3, 4, 5, and 6, the strain ARK1 is closely related to the ArkDPI strain, while the Mex-20 strain is grouped with other Mexican strains. In the phylogeny based on gene 2, all GI-9 strains are clustered together.

**GI-13 (4/91-strain like).** Phylogenetic analysis of all genes indicates that Mex-15 is closely related to the 4/91 vaccine and other Mexican GI-13 strains.

**GI-30.** We obtained four genomes from GI-30. The phylogenetic analyses of all genes clustered three GI-30 genomes (Mex-07-1, Mex-07-2, and Mex-07-3). However, the Mex-1 genome showed clustering depending on the analyzed gene. A recombination assay in the RDP4 program revealed that Mex-1 is a recombinant strain with a recombination breakpoint at position 8142 of the ORF 1a. The 5′ region of Mex-1 shares a high degree of similarity with GI-13 strains (98.5% with 4/91 vaccine), while the 3′ region displays a 99% similarity with a GI-9 Mexican strain (OM912694). Additionally, Mex-1 has another recombination breakpoint at position 24807 (beginning of gene 5); genes 5 and 6 exhibit high similarity with Massachusetts strains (99.5% with the H10 vaccine).

**GVIII-1.** Both strains of the GVIII genotype (Mex-12 and Mex-3009) are closely related but differ from other lineages in the phylogenies of ORF 1b and genes 2 to 6. However, the two GVIII strains exhibit lower similarity in the ORF 1a.

**GIX-1.** Both strains of the GIX genotype are closely related but distinct from other lineages in the gene 2 phylogeny. However, these strains do not form a cluster in the rest of the phylogenetic analyses. According to the RDP4 program, the strain Mex-56-7 was found to have three recombination breakpoints at positions 19525 (ORF1ab), 24518 (ORF M), and 26245 (ORF N) that divide the genome into four regions. The first and last regions are similar to the GI-3 strains sequenced here, while the second region corresponds to the S gene characteristic of the GIX genotype; the third region shares high similarity with the ArkDPI strain (GI-9). The Mex-14 strain is similar to GVIII-1 strains in genes 1 and 6, while the remaining genes are distinct from other lineages.

## 4. Discussion

The current IBV classification scheme based on whole-length S1 sequencing unified the categorization criteria and facilitated strain comparisons by determining genotypes and lineages circulating in different geographic regions [17]. By S1 analysis, Mexican strains were straightforwardly classified into eight lineages of the GI, GVIII, and GIX genotypes (GI-1, GI-3, GI-9, GI-13, GI-17, GI-30, GVIII-1, and GIX-1). Most of these lineages have persisted and cocirculated in Mexico for several years, indicating significant genomic variability in IBV (Figure 3). In contrast, a lineage is usually replaced in the United States after a particular time [26]. Similarly, in South America, most strains belonged to GI-11 and GI-16, with countries mostly having a single lineage that presumably replaced previous ones [27]. The recent widespread circulation of the GI-23 lineage in Brazil, which replaced the previously prevalent GI-11 lineage, provided further insights into the replacement hypothesis [28].

The analysis of S1 lineages provides valuable information, but studying the rest of the genome can reveal a distinct evolutionary story due to recombination and varying rates of evolution in certain regions. This background genome is sometimes a mosaic comprising horizontally transferred regions from other lineages as defined by complete S1 clustering. For this reason, obtaining the complete genome of circulating lineages is a further step toward understanding the intricate interplay of history and ongoing ecological and demographic processes that have resulted in the current distribution and diversity of IBV. Furthermore, it provides the scientific community with new insights into the complex microevolution of coronaviruses and their remarkable adaptability and spread. 

We utilized Illumina deep sequencing and bioinformatic analyses to compile fourteen completed genomes of IBV collected in Mexico over the past two decades. As a result, we obtained the complete genomes of new strains of GI-3, GI-9, and GI-13 and, for the first time, genomes of the indigenous Mexican lineages GI-30, GVIII-1, and GIX-1. 

Here is a brief overview of the key features of Mexican lineages, encompassing their genome variability within and between groups.

### 4.1. Mexican Lineages from Genotype 1

**GI-3 (JMK/Gray).** The respiratory and nephropathogenic GI-3 lineage was discovered in the USA during the 1960s and the late 1990s before being identified in Taiwan in 2006. Initially named JMK or the Gray serotype, the lineage comprises two similar viruses with different pathogenicity. Reference strains JMK (L14070) and Gray (L14069) were used to distinguish the two; the Gray variant is nephropathogenic, and the JMK virus is respirotropic [29]. Partial S1 sequences of this lineage were first reported in Mexico in 1998 [30], and this lineage continued circulating until the present [3,22]. While two Mexican genomes of GI-3 strains are available, they differ from the three Mexican GI-3 strains analyzed here and from the reference GI-3 genomes. 

**GI-9 (Ark).** The GI-9 lineage comprises vaccine and virulent field strains, commonly called Arkansas (Ark). In 1973, it emerged in Arkansas and was deemed genetically distinct from all other IBV serotypes recognized and referred to as Ark99 [31]. ArkDPI remained the most detected strain in the USA for decades, but its detection has since decreased due to the shift from ArkDPI to GA08 vaccine usage in the broiler sector [17,32]. In Western Europe, GI-9-like strains were only identified in flocks that received a commercial vaccine containing Ark in its formulation [33]. One of the GI-9 genomes obtained here has a high identity with the ArkDPI strain, while the other genome clustered with the eight GI-9 genomes obtained from south and central Mexico [22]. It is possible that this Ark-like genome came from a field strain because attenuated vaccines of the Arkansas serotype are not approved in Mexico. Distinguishing between the vaccine and field-origin strains poses a significant challenge. The Arkansas type is susceptible to genetic drift, which could trigger the emergence of significant variation within the lineage [34].

**GI-13 (4/91-like).** The GI-13 lineage, also known as 793B, 4/91, and CR88, has been identified in various regions worldwide [35]. It was first discovered in Europe during the 1980s and 1990s and was linked to severe respiratory syndromes [35,36]. The prototype strain of this lineage, which was isolated in Morocco in 1983, showed a 96% sequence similarity to the 793B variant, suggesting that the North African virus is the ancestor of the lineage [2]. In many countries, the IBV 793/B-type and the Massachusetts-based vaccines have been widely used for a long time. However, genetic analysis has shown that most detected 793/B field strains were vaccines or vaccine-derived [37]. GI-13 has been reported in Brazil, Chile, and Honduras. 

Serological evidence shows that GI-13 circulated in Mexico during the 1990s [35]. Recently, the 4/91 vaccine has been introduced in Mexico. On the other hand, recent studies have sequenced the genomes of 13 field strains and the 4/91 vaccine strain 1619/19 in broilers from central, north, and south Mexico [22]. We obtained a genome from a backyard chicken (fighting cook) clustered with most GI-13 strains and 4/91 vaccine. This suggests that GI-13 is present not only in commercial settings but also in domestic environments without a notorious intra-lineage genome divergence. 

Therefore, it is crucial to continuously monitor the prevalence and evolution of the 4/91 type to ensure maximum protection and application by attenuated vaccines. 

It is important to be cautious when introducing 4/91 vaccines, as strains of vaccine origin may persistently circulate and evolve in Mexico, acquiring behavior similar to field strains. Vaccination with a homologous strain can reduce the incidence of infection bronchitis disease, but it also increases the risk of recombination events with field strains that may produce new serotypes [37]. Our research findings require further evaluation but highlight the significance of continuously monitoring the prevalence and evolution of the 4/91 type to maximize protection and application by attenuated vaccines. 

**GI-30.** The GI-30 is a distinct group of Mexican origin that has persisted in central and western Mexico for more than 20 years (Table 1, Figure 3). No comparable S1 sequences are in the GenBank database (similarity < 85%), indicating that GI-30 has undergone unique divergence and local evolution. We obtained four genomes from this new lineage, with three showing close relatedness but divergence from other lineages. Phylogenetic analyses have revealed that the three related GI-30 strains share some genome regions with the GI-3 strains, which could suggest a common origin or an undetected ancestral recombinational event. On the other hand, the fourth GI-30 genome (Mex-1) is quite divergent due to extensive recombination involving the H120, 4/91, and GI-9 Mexican strains. Our findings support that GI-30 is an indigenous lineage that has undergone significant local differentiation in Mexico. 

### 4.2. Genomes of New Mexican Genotypes

**GVIII-1.** This genotype consists of strains gathered in Jalisco during 2020–2021 and the unique 98-0748 Mexican variant (AF288467 or UNAM-97) described by [17]. Strains classified under this genotype were previously identified through partial S1 sequencing and PCR-RFLP [30,38]. These original strains were collected from broilers with respiratory symptoms between 1997–1999 in the central Mexican states of Queretaro, Guanajuato, and San Luis Potosi. Serological studies of these strains also revealed antigenically different from Mass, Conn, and Ark serotypes [38]. This lineage should be considered indigenous, including samples collected 24 years apart, originally from central Mexican strains and the western state of Jalisco (Table 1). This unique lineage originated in Mexico and is likely associated with regional dissemination. This type is genetically and serologically divergent; therefore, the currently applied vaccines probably do not provide sufficient protection against GVIII-1.

The Mex-3009 and Mex-12 genomes have a high degree of similarity for most of their genes but differ from other IBV lineages. However, both GVIII strains vary in the ORF1a gene, possibly due to the high positive selection in the nsp3 region [39,40]. 

**GIX-1.** The GIX genotype includes two strains (Mex-14P and Mex-56-7) collected 13 years apart. Upon analysis of the S gene, these strains significantly differed from the closest lineage group. It is possible that this divergence occurred due to immunization pressure and natural selection in Mexico. IBV is known for its high genomic mutation and recombination rate, which can lead to the emergence of new genotypic groups under intense vaccination [41]. Both strains show evidence of recombination events in their genomes. The Mex-56-7 strain acquired gene sequences from Mexican GI-3 strains and the ArkDPI vaccine, while the ARK-14 strain obtained genomic regions from GVIII strains.

## 5. Conclusions

Mexico has a high genetic variability in IBV, possibly due to a combination of factors, including the poultry industry’s reach, the persistence of indigenous and foreign lineages, strains from nearby countries, and the backyard fowl industry’s relation to wild birds. Moreover, Mexico has been utilizing commercial vaccines that contain exotic IBV live strains that can circulate and recombine with field strains, resulting in a more diverse population. In contrast, other countries utilize epidemiological data and the cross-protection offered by the existing vaccines to handle viral variants in the field. Updating attenuated strains based on knowledge of circulating strains can be beneficial in preventing and controlling infectious bronchitis through long-term molecular epidemiological surveillance. 

## Figures and Tables

**Figure 1 viruses-15-01581-f001:**
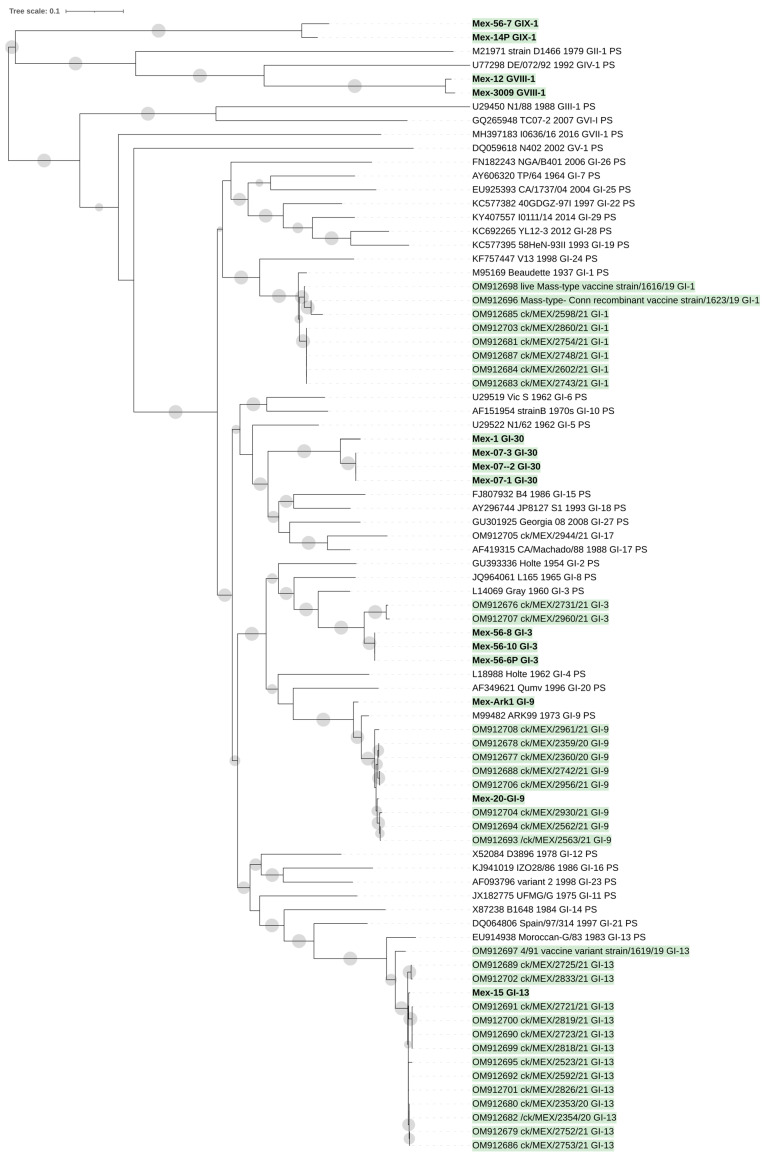
The phylogenetic tree was obtained with the maximum-likelihood method and GTR model with gamma distribution and invariant sites. Phylogenetic reconstruction was carried out using complete S1 sequences of the genomes here obtained (*n* = 14), other Mexican genomes available (*n* = 33), prototype IBV strains (*n* = 32) [17], and additional viruses from recently identified lineages and genotypes (*n* = 3). The Mexican strains are indicated with a green background, and the strains sequenced in this study are shown in bold. The gray circles represent node support values; the cutoff value is 50%.

**Figure 2 viruses-15-01581-f002:**
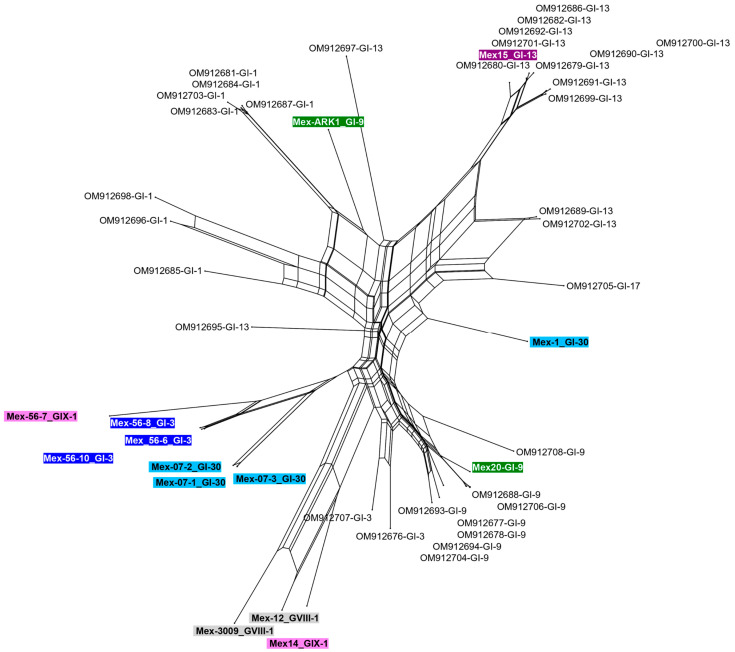
Reticulate network using Mexican complete genome sequences constructed with SplitsTree program. The Mexican strains sequenced in this study are highlighted. A color was assigned to each lineage.

**Figure 3 viruses-15-01581-f003:**
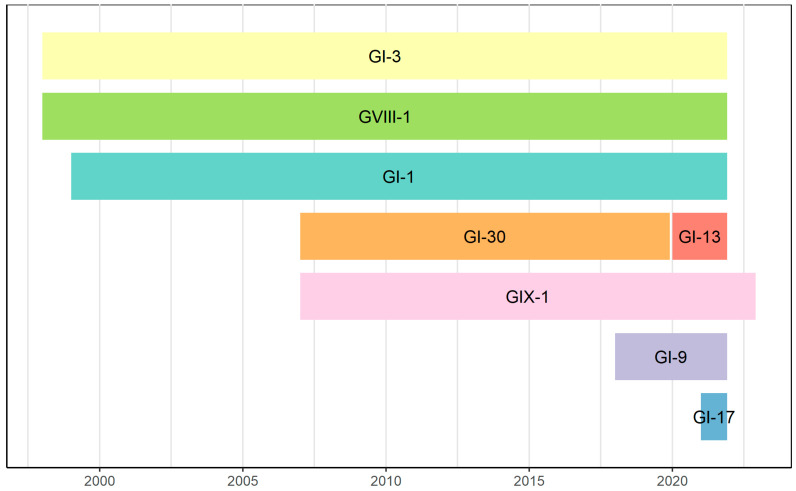
Temporal circulation in Mexico of eight lineages of the GI, GVIII, and GIX genotypes (GI-1, GI-3, GI-9, GI-13, GI-17, GI-30, GVIII-1, and GIX-1). The lineages are detailed with different colors, and the years in which sequence reports are available are indicated.

**Table 1 viruses-15-01581-t001:** Genome-sequenced strains from Mexico. Name, accession number, classification (genotype_lineage), bird type, collection year, and geographic origin are indicated from Mexican strains.

Strain	Accession	Classification	Type	Year	Origin
Mex-56-10	OR268739	GI-3	Hen	2007	Central Mexico
Mex-56-6	OR268740	GI-3	Hen	2007	Central Mexico
Mex-56-8	OR268741	GI-3	Hen	2007	Central Mexico
Mex-20	OR268742	GI-9	Broiler	2019	Western Mexico
Mex-Ark1	OR268743	GI-9	Hen	2018	Western Mexico
Mex-15	OR268744	GI-13	Hen	2020	Western Mexico
Mex-1	OR268745	GI-30	Backyard chicken	2019	Western Mexico
Mex-07-1	OR268746	GI-30	Hen	2007	Central Mexico
Mex-07-2	OR268747	GI-30	Hen	2007	Central Mexico
Mex-07-3	OR268748	GI-30	Hen	2007	Central Mexico
Mex-12	OR268751	GVIII-1	Hen	2020	Western Mexico
Mex-3009	OR268752	GVIII-1	NA	2021	Western Mexico
Mex-14P	OR268749	GIX-1	Hen	2020	Western Mexico
Mex-56-7	OR268750	GIX-1	NA	2007	Central Mexico

## Data Availability

All sequence data generated in this study are available in the GenBank database (Accession numbers are listed in Table 1).

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
