# Peer review of "Genome Variability of Infectious Bronchitis Virus in Mexico: High Lineage Diversity and Recurrent Recombination"

_viruses, 2023, doi:10.3390/v15071581_

Round 1

Reviewer 1 Report

The manuscript "Genome variability of infectious bronchitis virus in Mexico: high lineage diversity and recurrent recombination" is interesting and offers both new information and review of the state of art of IBV epidemiology in Mexico.

There are some points that could enrich the paper:

-the selection of the strains used for the study is not clear, no history about the sampled animals (clinical signs, vaccination, ...);

-if the strains were obtained from archive samples there is no need to describe the handling of the animals; otherwise if it was an active sampling it needs to be addressed with a completely different approach;

-there is little information about Mexican vaccination strategies, whose knowledge might actually help interpreting epidemiology;

-line 119: the reference for Marandino et al. (2017) is lacking

English writing can be improved:

line 26: variables

line 63: vaccination control challenging

lines 107-108: Chickens were euthanized by cervical dislocation and then examined the anatomical changes

line 113: stored isolated collected

line 124: library prep

line 204: gen

and other typos along the manuscript.

Author Response

The responses for reviewer 1 are in the attached file.

Reviewer 2 Report

The manuscript describes a genomic analysis on IBV strains having circulated in Mexico between 2007 and 2020. These analysis was first focused on S1 for a phylogeny study based on Valastro 2016 classification and then on the full genome to identify recombination events. The author described 14 genomes belonging to different genotypes/lineages, including specific ones to Mexico.

The manuscript is well constructed, synthesized and clear.

I have few suggestions to improve clarity among others:

-          Line 39: “It is the most important and best-studied Gammacoronavirus and the genus’ prototype and is currently present worldwide in industrial and backyard chickens [1–3].” > Too many "and". Sentence to modify

-          Line 60: ”the variant strains” > delete “the”

-          Line 107: “World Organization for Animal Health (OIE) animal welfare guidelines.” > replace OIE by the updated name : World Organisation for Animal Health

-          Line 122: “and converted to double-stranded cDNA with Maxima H Minus kit” > using which primers?

-          Line 158 – Table1 > These sequences has to be submitted to a genebank before article submission to obtain accession number. Please, add these numbers to the table. Indeed, the column was created for that purpose, with it is empty.

-          Line 167 – Figure 1 > Incomplete legend: colored samples, bold samples, grey round were not explained. In addition, bootstrap values are necessary (with a mention of the cutoff)

-          Line 187 – Figure 2 > Again, legend is missing. Explain the different colors.

-          Lines 198, 202, 205 : “(JMK/Gray), (Ark), (4/91-like) > Please, add notions/words to understand a bit more to what it is referred (eg strains-like)

-          Line 204: “based on gen 2” > gene

-          Line 219: “Both strains of the GIX genotype are closely related” > I think that mention in brackets strains name will help as their name is mentioned later one by one

-          Line 222: “at positions 19525, 24518, and 26245” > Please, precise in which genes are these breakpoints

-          Line 225: “with the ArkDPI strain.” > Please, re-mention lineage here to help comprehension

-          Lines 233-234: “confirming the circulation in Mexico.” > This part of the sentence can be removed

-          Lines 237-242: “In contrast…” > It needs to be rephrased. If I clearly understood, generally in countries, a specific lineage is progressively replaced by another one. eg, in USA, lineage ?? by lineage ?? [27], in Brazil, with GI-11 replaced by GI-23 [29], or ?? countries -in South America- with ? by ? [28].

-          Line 244 – Figure 3 > Add something to visualize where the new sequences described here are in the graphic will help the reader to easily see the detection of new lineages or the detection of already circulationg lineages previously described.

-          Lines 247-248: “but studying the background genome (the remaining genome without the entire S1 region)” > I would not consider the rest of the S gene as "background" or "backbone". Please, simply rephrase as the rest of the genome

-          Line 267: “Reference strains L14070 and L14069” > add name of stains

-          Line 269-270: “Partial S1 sequences of this lineage were first isolated in Mexico in 1998 [31] and have continued circulating” > The word "sequences" is not appropriated with isolated and circulate. Add the word "strain"?

-          Line 277: “ArkDPI” > Sometimes ArkDPI is with a space and sometimes not. Please, harmonize.

-          Lines 281-283: “One of the GI-9 genomes obtained here has a high identity with the Ark DPI strain, while the other genome clustered with the eight GI-9 genomes obtained from south and center Mexico [23].” > Can you add elements of explanation on why these lineage was found in Mexico, eg. using Ark-like strain vaccine?

-          Lines 328-329: “These original strains were collected from broilers between 1997–1999 in the central Mexican states” > Could you add information on clinical signs induced by this genotype?

-          Line 341: “includes two strains” > Give names in brackets

-          Lines 355-356: “population. In contrast, other countries rely on epidemiological information and cross-protection provided by the available vaccines to manage viral variants in the field” > Please rephrase. Is it linked to the next sentence, which is a consequence of that sentence? I'm not sure to understand what you meant

Author Response

The responses for reviewer 2 are in the attached file.

Reviewer 3 Report

The authors in this manuscript present a good description of IBV in Mexico. They refer particularly to fourteen genomes and their analysis. The methods are described properly but these could be improved, particularly those that were used to explore recombinant events. 

The results section is clear and presented in detail. However, it would be recommended to review the discussion. There is a great amount of background information about each lineage but limited discussion. Some of this information correspond to the introduction or even to the results section.

Considering the  results presented in the manuscript related to recombination events and the implications that this have for the disease control, a major discussion should be given to this topic.

Is there some information about clinical and/or pathological records that could be analyzed under the light of the molecular characteristics of the viruses found in the study? . 

It is suggested to explore the role that the use of different vaccines could have in the results (the type of vaccine or the use of different vaccines is not mentioned in the paper)

Author Response

The responses for reviewer 3 are in the attached file.
